# PUMA: SECURE INFERENCE OF LLAMA-7B IN FIVE MINUTES

## ABSTRACT

With ChatGPT as a representative, tons of companies have began to provide services based on large Transformers models. However, using such a service inevitably leak users' prompts to the model provider. Previous studies have studied secure inference for Transformer models using secure multiparty computation (MPC), where model parameters and clients' prompts are kept secret. Despite this, these frameworks are still limited in terms of model performance, efficiency, and deployment. To address these limitations, we propose framework PUMA to enable fast and secure Transformer model inference. Our framework designs high quality approximations for expensive functions such as GeLU and softmax, and significantly reduce the cost of secure inference while preserving the model performance. Additionally, we design secure Embedding and LayerNorm procedures that faithfully implement the desired functionality without undermining the Transformer architecture. PUMA is about $2\times$ faster than the state-of-the-art framework MPCFORMER(ICLR 2023) and has similar accuracy as plaintext models without fine-tuning (which the previous works failed to achieve). PUMA can even evaluate LLaMA-7B in around 5 minutes to generate 1 token. To our best knowledge, this is the first time that a model with such a parameter size is able to be evaluated under MPC.

## 1 INTRODUCTION

Pre-trained Transformer models (Vaswani et al., 2017) have attracted much attentions for their high performance in practical tasks (Radford & Narasimhan, 2018; Zhuge et al., 2021) and been widely in Deep Learning as a Service (DLaaS) paradigm (Soifer et al., 2019). However, these services can raise privacy concerns, such as in the case of ChatGPT (Brown et al., 2020), which requires either users to reveal their private prompts to the service provider or the service provider to release their proprietary trained weights to users.

One solution to address the privacy concerns of Transformer models service is Secure Multi-Party Computation (MPC) (Shamir, 1979; Yao, 1986; Goldreich et al., 1987), which can keep data and model weights private during inference. (Hao et al., 2022; Li et al., 2023; Akimoto et al., 2023; Liang et al., 2023; Liu & Liu, 2023) have proposed various ways to support secure Transformer models inference, but these approaches still have one or several of the following drawbacks:

**High inference cost.** Non-linear functions like GeLU and softmax are challenge to design in MPC. (Hao et al., 2022) computes these non-linear functions in a faithful way. *e.g.*, they design GeLU using tanh based on general MPC exponentiation method proposed by (Rathee et al., 2021). But these general methods are quite expensive in terms of computation and communication, and only tested under small bitwidth (e.g. below 32).

**Retraining required.** To reduce the cost of non-linear functions, several works (Li et al., 2023; Akimoto et al., 2023; Liu & Liu, 2023) suggested to approximate GeLU and softmax using simpler functions like ReLU and quadratics. These functions are up to an order of magnitude cheaper in MPC, but would introduce utility loss to the Transformer model. As a result, they require an extra step of model retraining (fine-tuning). However, retraining is unfriendly for data-limited participants, and might not achieve satisfactory performance (Kumar et al., 2022).

**Incompatible architectures.** (Li et al., 2023; Liang et al., 2023) proposed to modify the architecture of Transformer models to further accelerate secure inference, *e.g.*, decompose the embedding proce-

dure or reorganize the linear layers. Worsely, (Li et al., 2023) does not support secure LayerNorm and simulated the costs using BatchNorm, resulting in incorrect secure inference results. These modifications are in conflicts with existing plaintext Transformer systems, and would lead to deployment obstacles.

To summarize, in the field of MPC Transformer inference, achieving both model performance and efficiency is challenging, and people may ask the following question:

*Could pre-trained large transformer models be securely and efficiently evaluated with similar accuracy as in plaintext, without further retraining ?*

To address this challenge, we propose the PUMA framework, which is a fast and accurate end-to-end secure Transformer inference framework. Our contributions can be summarized as follows:

- **New Approximations for Non-linear Functions.** We propose more accurate and faster approximations for the expensive non-linear functions (*e.g.*, GeLU and softmax) in Transformer models. Different from existing works, we design the approximations based on the specialized properties of these non-linear functions to achieve both accuracy and efficiency.

- **Faster and More Accurate Secure Inference.** We make extensive experiments on 6 transformer models and 4 datasets, the results show that PUMA's precision is similar to plaintext ones' and is about $2\times$ faster than MPCFORMER (note that MPCFORMER does not achieve similar precision as PUMA). PUMA can even evaluate LLaMA-7B in around 5 minutes to generate one word. To our best knowledge, this is the first time that such a large language model is able to be evaluated under MPC.

- **End-to-End Framework compatible with plaintext.** We design and implement all the layers required by Transformer (including the Embedding and LayerNorm layers that are missing in other works) in MPC. This allows us to load and securely evaluate the pre-trained plaintext Transfomer models (*e.g.* downloaded from Hugging face) easily. To our best knowledge, PUMA is the first open-sourced MPC solution[1] that supports accurate inference of pre-trained Transformer models without further modifications such as re-training.

**Organization.** We summarize the related work in § 2 and present the background in § 3. We give PUMA's high-level view and concrete design in § 4. We analyze the experimental results in § 5 and conclude this work in § 6.

## 2 RELATED WORK

Secure Multiparty Computation (MPC) (Yao, 1986; Goldreich et al., 1987) enables distrusted parties to jointly compute a function while keeping their inputs private, and secure deep learning inference using MPC has gained much attention due its high privacy protection. These works operate in a variety of models and architectures, including two-party setting (Mohassel & Zhang, 2017; Liu et al., 2017; Mishra et al., 2020; Huang et al., 2022; Patra et al., 2021; Rathee et al., 2020), three-party setting (Wagh et al., 2019; Mohassel & Rindal, 2018; Wagh et al., 2020; Kumar et al., 2019; Patra & Suresh, 2020; Tan et al., 2021; Dong et al., 2023), or four-party setting (Byali et al., 2020; Dalskov et al., 2021). However, most of these approaches only consider secure inference of convolutional/deep neural networks, and cannot be directly extended to support Transformer models. Recently several research works (Hao et al., 2022; Li et al., 2023; Akimoto et al., 2023; Liang et al., 2023; Liu & Liu, 2023) have proposed MPC-based secure inference solutions for Transformer models, but these approaches still have limitations in terms of model performance, efficiency, and deployment. Among these works, MPCFORMER (Li et al., 2023) is the only one that have been open-sourced, it is based on CrypTen (Knott et al., 2021) which is a three-party framework that uses a non-colluding third party to produce correlated randomness for the client and server. Also their three-party model with non-colluding assumption has the highest concrete efficiency among different MPC settings. So we mainly compare our proposed framework PUMA with MPCFORMER under the same three-party setting.

---

[1]https://anonymous.4open.science/r/puma_benchmarks-6A81

## 3 BACKGROUND

### 3.1 NOTATIONS

The main used notations are as follows: $P_i$ represents the $i$-th computing party, $i \in \{0, 1, 2\}$. The uppercase bold letter $\mathbf{X}$ is used for matrices, and the lowercase bold letter $\mathbf{x}$ denotes vectors. $\mathbf{x}[i]$ denotes the $i$-th element of vector $\mathbf{x}$, while lowercase letter $x$ is used for scalar values. $\mathbb{Z}_{2^\ell}$ denotes the discrete ring modulo $2^\ell$, $\mathbb{R}$ denotes real numbers. $[\![\cdot]\!]$ is used for 2-out-of-3 replicated secret sharing (Araki et al., 2016; Mohassel & Rindal, 2018).

### 3.2 TRANSFORMER MODEL

Transformer models have achieved remarkable success in language understanding (Radford & Narasimhan, 2018; Devlin et al., 2019; Yang et al., 2019; Touvron et al., 2023), vision understanding (Zhuge et al., 2021; Dong et al., 2022; Chen et al., 2021), and etc. Two popular variants are Bert (Bidirectional Encoder Representations from Transformers) (Devlin et al., 2019) and GPT (Generative Pre-Trained models) (Radford & Narasimhan, 2018). A Transformer model (Vaswani et al., 2017) mainly consists of Embedding, **Attention**, **Feed-Forward Network**, and **LayerNorm** sub-layers:

**Attention.** Given inputs $(\mathbf{Q}, \mathbf{K}, \mathbf{V})$, the Attention function is computed as $\mathsf{Attention}(\mathbf{Q}, \mathbf{K}, \mathbf{V}) = \mathsf{softmax}(\mathbf{Q} \cdot \mathbf{K}^\mathsf{T} + \mathbf{M}) \cdot \mathbf{V}$, where $\mathbf{M}$ can be viewed as a bias matrix. Besides, (Vaswani et al., 2017) proposed Multi-Head Attention to jointly attend to information from different representation subspaces at different positions.

**Feed-Forward Network** (FFN)**.** FFN is applied to each position separately and identically. This consists of two linear transformations with an activation in between, and the most commonly used activation function is GeLU. Given input $\mathbf{x}$ and parameters $\{\mathbf{W}_1, \mathbf{b}_1, \mathbf{W}_2, \mathbf{b}_2\}$, FFN can be formalized as $\mathsf{FFN}(\mathbf{x}) = \mathbf{W}_2 \mathsf{GeLU}(\mathbf{W}_1 \mathbf{x} + \mathbf{b}_1) + \mathbf{b}_2$. Note that the parameters of linear transformations are different from layer to layer.

**LayerNorm.** Given vector $\mathbf{x} \in \mathbb{R}^n$, LayerNorm is defined as: $\mathsf{LayerNorm}(\mathbf{x})[i] = \gamma \cdot \frac{\mathbf{x}[i] - \mu}{\sqrt{\sigma}} + \beta$, where $(\gamma, \beta)$ are trained parameters, $\mu = \frac{\sum_{i=1}^n \mathbf{x}[i]}{n}$, and $\sigma = \sum_{i=1}^n (\mathbf{x}[i] - \mu)^2$.

### 3.3 2-OUT-OF-3 REPLICATED SECRET SHARING

A secret value $x \in \mathbb{Z}_{2^\ell}$ is shared by three random values $x_0, x_1, x_2 \in \mathbb{Z}_{2^\ell}$ with $x = x_0 + x_1 + x_2 \pmod{2^\ell}$. In 2-out-of-3 replicated secret sharing (denoted as $[\![\cdot]\!]$-sharing), party $P_i$ gets $[\![x]\!]_i = (x_i, x_{i+1})$. Without special declaration, we compute in $\mathbb{Z}_{2^\ell}$ and omit $\pmod{2^\ell}$ for brevity. In the case of $\ell > 1$ (*e.g.*, $\ell = 64$) which support arithmetic operations (*e.g.*, $+$, $-$, and $\cdot$), we refer to this type as *Arithmetic Sharing* and use notation $[\![\cdot]\!]$. *Boolean Sharing* ($[\![\cdot]\!]^\mathsf{B}$) refers to $\ell = 1$ where $(+, -)$ and $\cdot$ are respectively replaced by bit-wise $\oplus$ and $\wedge$.

**Addition.** Let $(c_1, c_2, c_3)$ be public constants, and $([\![x]\!], [\![y]\!])$ be two secret-shared values. Then, $[\![c_1 x + c_2 y + c_3]\!]$ can be computed as $(c_1 x_0 + c_2 y_0 + c_3, c_1 x_1 + c_2 y_1, c_1 x_2 + c_2 y_2)$ where $P_i$ can compute its share locally. When $(c_1 = 1, c_2 = 1, c_3 = 0)$, we get $[\![x + y]\!]$.

**Multiplication.** In secure multiplication protocol $\Pi_{\mathsf{Mul}}$, given two shared values $[\![x]\!]$ and $[\![y]\!]$, parties follows steps: i) First, $P_i$ computes $z_i = x_i y_i + x_{i+1} y_i + x_i y_{i+1}$ locally, ii) Parties then perform *re-sharing* by letting $P_i$ sends $z_i' = \alpha_i + z_i$ to $P_{i-1}$, where $\alpha_0 + \alpha_1 + \alpha_2 = 0$ ($P_i$ can generate $\alpha_i$ in the setup phase as Mohassel & Rindal (2018)). iii) Finally, $\{(z_0', z_1'), (z_1', z_2'), (z_2', z_0')\}$ form $[\![x \cdot y]\!]$.

**Underlying Protocols.** In addition to addition and multiplication, PUMA relies on several other underlying protocols: boolean-arithmetic multiplication ($\Pi_{\mathsf{Mul_{BA}}}$), square $\Pi_{\mathsf{Square}}$, equality test ($\Pi_{\mathsf{Eq}}$), less than ($\Pi_{\mathsf{LT}}$), reciprocal ($\Pi_{\mathsf{Recip}}$), maximum ($\Pi_{\mathsf{Max}}$), and reciprocal of square root ($\Pi_{\mathsf{rSqrt}}$), from the state-of-the-art works. We employ them in a black-box manner, and only enumerate the inputs and outputs of these protocols as follows:

- $[\![z]\!] = \Pi_{\mathsf{Mul_{BA}}}([\![b]\!]^\mathsf{B}, [\![x]\!])$, s.t. $z = b \cdot x$
- $[\![z]\!]^\mathsf{B} = \Pi_{\mathsf{LT}}([\![x]\!], [\![y]\!])$, s.t. $z = 1\{x < y\}$
- $[\![z]\!] = \Pi_{\mathsf{Square}}([\![x]\!])$, s.t. $z = x^2$
- $[\![z]\!] = \Pi_{\mathsf{Recip}}([\![x]\!])$, s.t. $z = 1/x$
- $[\![z]\!]^\mathsf{B} = \Pi_{\mathsf{Eq}}([\![x]\!], [\![y]\!])$, s.t. $z = 1\{x = y\}$
- $[\![z]\!] = \Pi_{\mathsf{rSqrt}}([\![x]\!])$, s.t. $z = 1/\sqrt{x}$

- $[\![z]\!] = \Pi_{\mathsf{Max}}([\![\mathbf{x}]\!])$, s.t. $z = \mathsf{maximum}(\mathbf{x})$

$1\{e\}$ returns 1 that when condition $e$ is true, and $0$ otherwise. For detailed protocol constructions, please refer to (Mohassel & Rindal, 2018; Lu et al., 2020; Keller, 2020).

**Fixed-Point Representation & Truncation.** Real numbers has to be encoded into fixed-point numbers before represented in finite rings/fields. To avoid overflow, $\Pi_{\mathsf{Trunc}}^{f}$ has to be used after each fixed-point multiplication to truncate the least $f$ bits securely. For simpler description, we include $\Pi_{\mathsf{Trunc}}^{f}$ in $\Pi_{\mathsf{Mul}}$ and $\Pi_{\mathsf{Square}}$ by default and and do not explicitly mention it in our protocol designs.

The above operations can be easily extended to vectors and matrices, and we use the same notation for vector and matrix operations for simplicity. For more details, please refer to (Mohassel & Rindal, 2018; Wagh et al., 2020).

**Threat Model.** Following previous works (Mohassel & Rindal, 2018; Li et al., 2023), PUMA is secure against a semi-honest adversary that corrupts no more than one of the three computing parties. Semi-honest means such an adversary will follow the protocol specifications, but may try to learn other's private information during the protocol. Please note that PUMA cannot defend against attacks based on inference results, and the mitigation of such attacks (*e.g.*, differential privacy (Abadi et al., 2016)) falls outside the scope of this study.

## 4 SECURE DESIGN OF PUMA

In this section, we first present an overview of PUMA, and present the protocols for secure GeLU , $\mathrm{softmax}$, embedding, and LayerNorm used by PUMA. Note that the linear layers such as matrix multiplication are straightforward in replicated secret sharing, so we mainly describe our protocols for non-linear layers in this manuscript.

### 4.1 OVERVIEW OF PUMA

To achieve secure inference of Transformer models, PUMA defines three kinds of roles: one model owner, one client, and three computing parties. The model owner and the client provide their models or inputs to the computing parties (i.e., $P_0$, $P_1$, and $P_2$) in a secret-shared form, then the computing parties execute the MPC protocols and send the results back to the client. Note that the model owner and client can also act as one of the computing party, we describe them separately for generality. *e.g.*, when the model owner acts as $P_0$, the client acts as $P_1$, a third-party dealer acts as $P_2$, the system model becomes the same with MPCFORMER (Li et al., 2023).

During the secure inference process, a key invariant is maintained: For any layer, the computing parties always start with 2-out-of-3 replicated secret shares of the previous layer's output and the model weights, and end with 2-out-of-3 replicated secret shares of this layer's output. As the shares do not leak any information to each party, this ensures that the layers can be sequentially combined for arbitrary depths to obtain a secure computation scheme for any Transformer-based model.

### 4.2 PROTOCOL FOR SECURE GeLU

Most of the current approaches view the GeLU function as a composition of smaller functions and try to optimize each piece of them, making them to miss the chance of optimizing the private GeLU as a whole. Given the GeLU function:

$$\mathsf{GeLU}(x) = \frac{x}{2} \cdot \left( 1 + \tanh \left( \sqrt{\frac{2}{\pi}} \cdot \left( x + 0.044715 \cdot x^3 \right) \right) \right), \tag{1}$$
$$\approx x \cdot \mathsf{sigmoid}(0.071355 \cdot x^3 + 1.595769 \cdot x)$$

these approaches (Hao et al., 2022; Wang et al., 2022) focus either on designing approximate protocols for function $\tanh$ or using existing general MPC protocols of exponentiation and reciprocal for $\mathsf{sigmoid}$.

However, none of current approaches have utilized the fact that GeLU function is almost linear on the two sides (*i.e.*, $\mathsf{GeLU}(x) \approx 0$ for $x < -4$ and $\mathsf{GeLU}(x) \approx x$ for $x > 3$). Within the short

---

**Algorithm 1** Secure GeLU Protocol $\Pi_{\mathsf{GeLU}}$

---

**Input:** $P_i$ holds the 2-out-of-3 replicate secret share $[\![x]\!]_i$ for $i \in \{0, 1, 2\}$
**Output:** $P_i$ gets the 2-out-of-3 replicate secret share $[\![y]\!]_i$ for $i \in \{0, 1, 2\}$, where $y = \mathsf{GeLU}(x)$.
1: $P_0$, $P_1$, and $P_2$ jointly compute

$$
\begin{aligned}
[\![b_0]\!]^{\mathsf{B}} &= \Pi_{\mathsf{LT}}([\![x]\!], -4), &&\triangleright b_0 = 1\{x < -4\} \\
[\![b_1]\!]^{\mathsf{B}} &= \Pi_{\mathsf{LT}}([\![x]\!], -1.95), &&\triangleright b_1 = 1\{x < -1.95\} \\
[\![b_2]\!]^{\mathsf{B}} &= \Pi_{\mathsf{LT}}(3, [\![x]\!]), &&\triangleright b_2 = 1\{3 < x\}
\end{aligned}
$$

and compute $[\![z_0]\!]^{\mathsf{B}} = [\![b_0]\!]^{\mathsf{B}} \oplus [\![b_1]\!]^{\mathsf{B}}$, $[\![z_1]\!]^{\mathsf{B}} = [\![b_1]\!]^{\mathsf{B}} \oplus [\![b_2]\!]^{\mathsf{B}} \oplus 1$, and $[\![z_2]\!]^{\mathsf{B}} = [\![b_2]\!]^{\mathsf{B}}$. Note that $z_0 = 1\{-4 \leq x < -1.95\}$, $z_1 = 1\{-1.95 \leq x \leq 3\}$, and $z_2 = 1\{x > 3\}$.
2: Jointly compute $[\![x^2]\!] = \Pi_{\mathsf{Square}}([\![x]\!])$, $[\![x^3]\!] = \Pi_{\mathsf{Mul}}([\![x]\!], [\![x^2]\!])$, $[\![x^4]\!] = \Pi_{\mathsf{Square}}([\![x^2]\!])$, and $[\![x^6]\!] = \Pi_{\mathsf{Square}}([\![x^3]\!])$.
3: Computing polynomials $[\![F_0(x)]\!]$ and $[\![F_1(x)]\!]$ based on $\{[\![x]\!], [\![x^2]\!], [\![x^3]\!], [\![x^4]\!], [\![x^6]\!]\}$ as equation (2) securely.
4: **return** $[\![y]\!] = \Pi_{\mathsf{Mul_{BA}}}([\![z_0]\!]^{\mathsf{B}}, [\![F_0(x)]\!]) + \Pi_{\mathsf{Mul_{BA}}}([\![z_1]\!]^{\mathsf{B}}, [\![F_1(x)]\!]) + \Pi_{\mathsf{Mul_{BA}}}([\![z_2]\!]^{\mathsf{B}}, [\![x]\!])$.

---

interval $[-4, 3]$ of GeLU, we suggest a piece-wise approximation of low-degree polynomials is a more efficient and easy-to-implement choice for its secure protocol. Concretely, our piece-wise low-degree polynomials are shown as equation (2):

$$
\mathsf{GeLU}(x) = \begin{cases}
0, & x < -4 \\
F_0(x), & -4 \leq x < -1.95 \\
F_1(x), & -1.95 \leq x \leq 3 \\
x, & x > 3
\end{cases}, \tag{2}
$$

where polynomials $F_0()$ and $F_1()$ are computed by library numpy.ployfit[2] as equation (3). Surprsingly, the above simple poly fit works very well and our max error $< 0.01403$, median error $< 4.41e - 05$, and mean error $< 0.00168$.

$$
\begin{cases}
F_0(x) &= -0.011034134030615728x^3 - 0.11807612951181953x^2 \\
&\quad -0.42226581151983866x - 0.5054031199708174 \\
F_1(x) &= 0.0018067462606141187x^6 - 0.037688200365904236x^4 \\
&\quad +0.3603292692789629x^2 + 0.5x + 0.008526321541038084
\end{cases} \tag{3}
$$

Formally, given secret input $[\![x]\!]$, our secure GeLU protocol $\Pi_{\mathsf{GeLU}}$ is constructed as algorithm 1.

### 4.3 PROTOCOL FOR SECURE SOFTMAX

In the function $\mathrm{Attention}(\mathbf{Q}, \mathbf{K}, \mathbf{V}) = \mathrm{softmax}(\mathbf{Q} \cdot \mathbf{K}^{\mathsf{T}} + \mathbf{M}) \cdot \mathbf{V}$, the key challenge is computing function $\mathrm{softmax}$. For the sake of numerical stability, the $\mathrm{softmax}$ function is computed as

$$
\mathrm{softmax}(\mathbf{x})[i] = \frac{\exp(\mathbf{x}[i] - \bar{x} - \epsilon)}{\sum_i \exp(\mathbf{x}[i] - \bar{x} - \epsilon)}, \tag{4}
$$

where $\bar{x}$ is the maximum element of the input vector $\mathbf{x}$. For the normal plaintext softmax, $\epsilon = 0$. For a two-dimension matrix, we apply equation (4) to each of its row vector.

Formally, our detailed secure protocol $\Pi_{\mathrm{softmax}}$ is illustrated in algorithm 2, where we propose two optimizations:

- For the first optimization, we set $\epsilon$ in equation 4 to a tiny and positive value, e.g., $\epsilon = 10^{-6}$, so that the inputs to exponentiation in equation 4 are all negative. We exploit the negative operands for acceleration. Particularly, we compute the exponentiation using the Taylor series (Tan et al., 2021) with a simple clipping

$$
\mathsf{negExp}(x) = \begin{cases}
0, & x < T_{\exp} \\
(1 + \frac{x}{2^t})^{2^t}, & x \in [T_{\exp}, 0].
\end{cases} \tag{5}
$$

---
[2]https://numpy.org/doc/stable/reference/generated/numpy.polyfit.html

---

**Algorithm 2** Secure softmax Protocol $\Pi_{\text{softmax}}$

---

**Input:** $P_i$ holds the replicate secret share $[\![\mathbf{x}]\!]_i$ for $i \in \{0, 1, 2\}$, and $\mathbf{x}$ is a vector of size $n$.
**Output:** $P_i$ gets the replicate secret share $[\![\mathbf{y}]\!]_i$ for $i \in \{0, 1, 2\}$, where $\mathbf{y} = \text{softmax}(\mathbf{x})$.
1: $P_0$, $P_1$, and $P_2$ jointly compute $[\![\mathbf{b}]\!]^\mathsf{B} = \Pi_{\text{LT}}(T_{\exp}, [\![\mathbf{x}]\!])$ and the maximum $[\![\bar{x}]\!] = \Pi_{\text{Max}}([\![\mathbf{x}]\!])$.
2: Parties locally computes $[\![\hat{\mathbf{x}}]\!] = [\![\mathbf{x}]\!] - [\![\bar{x}]\!] - \epsilon$, and jointly compute $[\![\mathbf{z}_0]\!] = 1 + \Pi_{\text{Trunc}}^t([\![\hat{\mathbf{x}}]\!])$.
3: **for** $j = 1, 2, \ldots, t$ **do**
4: $\quad [\![\mathbf{z}_j]\!] = \Pi_{\text{Square}}([\![\mathbf{z}_{j-1}]\!])$.
5: **end for**
6: Parties locally compute $[\![z]\!] = \sum_{i=1}^n [\![\mathbf{z}[i]]\!]$ and jointly compute $[\![1/z]\!] = \Pi_{\text{Recip}}([\![z]\!])$.
7: Parties jointly compute $[\![\mathbf{z}/z]\!] = \Pi_{\text{Mul}}([\![\mathbf{z}]\!], [\![1/z]\!])$
8: **return** $[\![\mathbf{y}]\!] = \Pi_{\text{Mul}_{\text{BA}}}([\![\mathbf{b}]\!]^\mathsf{B}, [\![\mathbf{z}/z]\!])$.

---

Indeed, we apply the less-than for the branch $x < T_{\exp}$ The division by $2^t$ can be achieved using $\Pi_{\text{Trunc}}^t$ since the input is already negative. Also, we can compute the power-of-$2^t$ using $t$-step sequences of square function $\Pi_{\text{square}}$ and $\Pi_{\text{Trunc}}^f$. Suppose our MPC program uses 18-bit fixed-point precision. Then we set $T_{\exp} = -14$ given $\exp(-14) < 2^{-18}$, and empirically set $t = 5$.

- Our second optimization is to reduce the number of divisions, which ultimately saves computation and communication costs. To achieve this, for a vector $\mathbf{x}$ of size $n$, we have replaced the operation $\text{Div}(\mathbf{x}, \text{Broadcast}(y))$ with $\mathbf{x} \cdot \text{Broadcast}(\frac{1}{y})$, where $y = \sum_{i=1}^n \mathbf{x}[i]$. By making this replacement, we effectively reduce $n$ divisions to just one reciprocal operation and $n$ multiplications. This optimization is particularly beneficial in the case of the $\text{softmax}$ operation. The $\frac{1}{y}$ in the $\text{softmax}$ operation is still large enough to maintain sufficient accuracy under fixed-point values. As a result, this optimization can significantly reduce the computational and communication costs while still providing accurate results.

## 4.4 PROTOCOL FOR SECURE EMBEDDING

The current secure embedding procedure described in (Li et al., 2023) necessitates the client to generate a one-hot vector using the token id locally. This deviates from a plaintext Transformer workflow where the one-hot vector is generated inside the model. As a result, they have to carefully strip off the one-hot step from the pre-trained models, and add the step to the client side, which could be an obstacle for deployment.

To address this issue, we propose a secure embedding design as follows. Assuming that the token id $\in [n]$ and all embedding vectors are denoted by $\mathbb{E} = (\mathbf{e}_1^T, \mathbf{e}_2^T, \ldots, \mathbf{e}_n^T)$, the embedding can be formulated as $\mathbf{e}_{\text{id}} = \mathbf{E}[\text{id}]$. Given $(\text{id}, \mathbb{E})$ are in secret-shared fashion, our secure embedding protocol $\Pi_{\text{Embed}}$ works as follows:

- The computing parties securely compute the one-hot vector $[\![\mathbf{o}]\!]^\mathsf{B}$ after receiving $[\![\text{id}]\!]$ from the client. Specifically, $[\![\mathbf{o}[i]]\!]^\mathsf{B} = \Pi_{\text{Eq}}(i, [\![\text{id}]\!])$ for $i \in [n]$.

- The parties can compute the embedded vector via $[\![\mathbf{e}_{\text{id}}]\!] = \Pi_{\text{Mul}_{\text{BA}}}([\![\mathbb{E}]\!], [\![\mathbf{o}]\!]^\mathsf{B})$, where does not require secure truncation.

In this way, our $\Pi_{\text{Embed}}$ does not require explicit modification of the workflow of plaintext Transformer models, at the cost of more $\Pi_{\text{Eq}}$ and $\Pi_{\text{Mul}_{\text{BA}}}$ operations.

## 4.5 PROTOCOL FOR SECURE LAYERNORM

Recall that given a vector $\mathbf{x}$ of size $n$, $\text{LayerNorm}(\mathbf{x})[i] = \gamma \cdot \frac{\mathbf{x}[i] - \mu}{\sqrt{\sigma}} + \beta$, where $(\gamma, \beta)$ are trained parameters, $\mu = \frac{\sum_{i=1}^n \mathbf{x}[i]}{n}$, and $\sigma = \sum_{i=1}^n (\mathbf{x}[i] - \mu)^2$. In MPC, the key challenge is the evaluation of the divide-square-root $\frac{\mathbf{x}[i] - \mu}{\sqrt{\sigma}}$ formula. To securely evaluate this formula, CrypTen sequentially executes the MPC protocols of square-root, reciprocal, and multiplication. However, we observe that $\frac{\mathbf{x}[i] - \mu}{\sqrt{\sigma}}$ is equal to $(\mathbf{x}[i] - \mu) \cdot \sigma^{-1/2}$. And in the MPC side, the costs of computing the inverse-square-root $\sigma^{-1/2}$ is similar to that of the square-root operation (Lu et al., 2020). Besides, inspired by the

---

**Algorithm 3** Secure LayerNorm Protocol $\Pi_{\mathsf{LayerNorm}}$

---

**Input:** $P_i$ holds the replicate secret share $[\![\mathbf{x}]\!]_i$ for $i \in \{0, 1, 2\}$, and $\mathbf{x}$ is a vector of size $n$.
**Output:** $P_i$ gets the replicate secret share $[\![\mathbf{y}]\!]_i$ for $i \in \{0, 1, 2\}$, where $\mathbf{y} = \mathsf{LayerNorm}(\mathbf{x})$.
1: $P_0$, $P_1$, and $P_2$ compute $[\![\mu]\!] = \frac{1}{n} \cdot \sum_{i=1}^{n} [\![\mathbf{x}[i]]\!]$ and $[\![\sigma]\!] = \sum_{i=1}^{n} \Pi_{\mathsf{Square}}([\![\mathbf{x}]\!] - [\![\mu]\!])[i]$.
2: Parties jointly compute $[\![\sigma^{-1/2}]\!] = \Pi_{\mathsf{rSqrt}}([\![\sigma]\!])$.
3: Parties jointly compute $[\![\mathbf{c}]\!] = \Pi_{\mathsf{Mul}}(([\![\mathbf{x}]\!] - [\![\mu]\!]), [\![\sigma^{-1/2}]\!])$
4: **return** $[\![\mathbf{y}]\!] = \Pi_{\mathsf{Mul}}([\![\gamma]\!], [\![\mathbf{c}]\!]) + [\![\beta]\!]$.

---

second optimization of § 4.3, we can first compute $\sigma^{-1/2}$ and then $\mathsf{Broadcast}(\sigma^{-1/2})$ to support fast and secure $\mathsf{LayerNorm}(\mathbf{x})$. And our formal protocol $\Pi_{\mathsf{LayerNorm}}$ is shown in algorithm 3.

## 5 EXPERIMENTAL EVALUATIONS

**Implementation.** We implement PUMA on top of SecretFlow-SPU (Ma et al., 2023) in C++ and Python. We encode the data in a fixed-point form under ring $\mathbb{Z}_{2^{64}}$ with 18-bit fractional part. Our experiments are ran on 3 Alibaba Cloud ecs.g7.8xlarge servers with 32 vCPUs and 128GB RAM each. The CPU model is Intel Xeon(Ice Lake) Platinum 8369B CPU @ 2.70GHz. We evaluate PUMA on Ubuntu 20.04.6 LTS with Linux kernel 5.4.0-144-generic. Our bandwidth is about 5Gbps and round trip time is about 1ms.

**Models & Datasets.** We evaluate PUMA on seven NLP models: Bert-Base, Roberta-Base, and Bert-Large (Devlin et al., 2019); GPT2-Base, GPT2-Medium, and GPT2-Large (Radford & Narasimhan, 2018); and LLaMA-7B (Touvron et al., 2023). We measure the Bert performance for three NLP tasks over the datasets of Corpus of Linguistic Acceptability (CoLA), Recognizing Textual Entailment (RTE), Stanford Question Answering Dataset (QNLI) from GLUE benchmarks (Wang et al., 2019), and GPT2 performance on Wikitext-103 V1 (Merity et al., 2016).

**Baseline.** We compare PUMA to the most similar prior work MPCFORMER (Li et al., 2023). But for fair comparison, we have the following considerations: i) As MPCFORMER neither supports loading pretrained transformer models nor implements LayerNorm faithfully[3], we cannot achieve meaningful secure inference results using their framework. Therefore, we compare our performance to that of plaintext (floating-point) to show our precision guarantee. ii) MPCFORMER with *Quad* approximations requires retraining the modified models. As PUMA does not require retraining, we compare our cost to that of MPCFORMER without *Quad* approximations. Also, we re-run MPCFORMER in our environment.

### 5.1 PRECISION

We randomly sample 500 instances from the validation split of each dataset within the GLUE benchmark to assess Bert inference performance, and for evaluating GPT2 inference performance, we select 100 samples from the validation split of Wikitext-103 V1. We compare our secure model inference performance to that of plaintext (floating-point) in Table 1 and 2 to show our precision guarantee.

In Table 1, we show the Matthews correlation/accuracy of plaintext and PUMA on the Bert models. We observe that the accuracy achieved by PUMA matches the accuracy of the plaintext Flax code. Specifically, the accuracy difference does not exceed 0.011 over all datasets. Moreover, in Table 2, we also compare our perplexity on dataset Wikitext-103 V1 with the plaintext baseline on GPT2 models. The results are similar and the perplexity differences do not exceed 0.02 over all models.

The above accuracy and perplexity advantages experimentally validate that our protocols are numerically precise.

---

[3]As MPCFORMER does not support loading pre-trained Transformer models, we did an experiment in plaintext Bert-Base that replaced LayerNorm with BatchNorm as MPCFORMER did. This resulted in a significant drop in the MCC score for CoLA task from 0.616 to $-0.020$. On the contrary, PUMA achieves an MCC score of 0.613.

Table 1: Performance on GLUE benchmark of Bert-Base, Roberta-Base, and Bert-Large on CoLA, RTE, and QNLI. Matthews correlation is reported for CoLA. Accuracy is reported for other datasets.

| Model | Bert-Base | | | Roberta-Base | | | Bert-Large | | |
|---|---|---|---|---|---|---|---|---|---|
| TASK | CoLA | RTE | QNLI | CoLA | RTE | QNLI | CoLA | RTE | QNLI |
| Plaintext | 0.616 | 0.700 | 0.916 | 0.629 | 0.805 | 0.920 | 0.686 | 0.755 | 0.922 |
| PUMA | 0.613 | 0.700 | 0.916 | 0.618 | 0.805 | 0.918 | 0.690 | 0.747 | 0.918 |

Table 2: Perplexity of GPT2-Base, GPT2-Medium, and GPT2-Large on Wikitext-103 V1.

| Model | GPT2-Base | GPT2-Medium | GPT2-Large |
|---|---|---|---|
| Plaintext | 16.284 | 12.536 | 10.142 |
| PUMA | 16.284 | 12.540 | 10.161 |

Table 3: Costs of Bert-Base, Roberta-Base, and Bert-Large for one sentence of length 128. Time is in seconds and Communication (Comm. for short) is in GB, which is the same for the following tables.

| Model | Bert-Base | | Roberta-Base | | Bert-Large | |
|---|---|---|---|---|---|---|
| Costs | Time | Comm. | Time | Comm. | Time | Comm. |
| MPCFORMER | 55.320 | 12.089 | 57.256 | 12.373 | 141.222 | 32.577 |
| PUMA | 33.913 | 10.773 | 41.641 | 11.463 | 73.720 | 27.246 |
| Improv. | $1.631\times$ | $1.122\times$ | $1.375\times$ | $1.079\times$ | $1.916\times$ | $1.195\times$ |

Table 4: Costs of GPT2-Base, GPT2-Medium, and GPT2-Large. The input sentence is of length 32, all of the costs are for generating 1 token.

| Model | GPT2-Base | | GPT2-Medium | | GPT2-Large | |
|---|---|---|---|---|---|---|
| Costs | Time | Comm. | Time | Comm. | Time | Comm. |
| MPCFORMER | 34.889 | 4.999 | 73.078 | 11.766 | 129.095 | 22.522 |
| PUMA | 15.506 | 3.774 | 30.272 | 7.059 | 54.154 | 11.952 |
| Improv. | $2.250\times$ | $1.325\times$ | $2.414\times$ | $1.667\times$ | $2.383\times$ | $1.884\times$ |

Table 5: Costs of Bert-Base and GPT2-Base for different input length (denoted as #Input). The input lengths for Bert-Base and GPT2-Base are respectively $\{64, 128, 256\}$ and $\{16, 32, 64\}$. GPT2-Base generates 1 token.

| | #Input | 64/16 | | 128/32 | | 256/64 | |
|---|---|---|---|---|---|---|---|
| | Costs | Time | Comm. | Time | Comm. | Time | Comm. |
| | MPCFORMER | 36.354 | 5.707 | 55.320 | 12.089 | 112.453 | 29.927 |
| Bert | PUMA | 21.141 | 4.881 | 33.913 | 10.773 | 61.210 | 26.004 |
| | Improv. | $1.720\times$ | $1.169\times$ | $1.631\times$ | $1.122\times$ | $1.837\times$ | $1.151\times$ |
| | MPCFORMER | 29.695 | 4.011 | 34.889 | 4.999 | 43.344 | 7.318 |
| GPT2 | PUMA | 11.056 | 1.875 | 15.506 | 3.777 | 24.860 | 7.821 |
| | Improv. | $2.686\times$ | $2.139\times$ | $2.250\times$ | $1.324\times$ | $1.744\times$ | $0.936\times$ |

## 5.2 INFERENCE COSTS

We compare PUMA's inference cost to that of MPCFORMER in Table 3 and Table 4. The costs are for processing one input sentence: i) For Bert models the input sentence is of length 128. ii) For GPT2 models the input length is 32 and generate 1 new word.

On the 3 Bert models in Table 3, PUMA is $1.375 \sim 1.916\times$ faster than MPCFORMER, and is $1.079 \sim 1.195\times$ more communication-efficient. For the GPT2 models in Table 4, PUMA is $2.250 \sim 2.414\times$ faster than MPCFORMER, and is $1.325 \sim 1.884\times$ more communication-efficient.

We observe that PUMA's improvements increase as the model size grows, particularly for the GPT2 models. This trend is because our specialized optimizations are more effective when processing large-scale evaluations.

## 5.3 SCALABILITY

In this subsection, we measure the costs of evaluating PUMA on Bert-Base and GPT2-Base models for batched inputs, varying-length inputs, and varying-length outputs (only for GPT2-Base). We also compare our costs to those of MPCFORMER to demonstrate our improvements.

**Input Length Evaluation.** Table 5 shows our costs on varying-length inputs, we evaluate Bert-Base on inputs of length $\{64, 128, 256\}$, and GPT2-Base on inputs of length $\{16, 32, 64\}$. For Bert-Base, PUMA is $1.631 \sim 1.837\times$ faster, and for GPT2-Base, PUMA is $1.744 \sim 2.686\times$ faster.

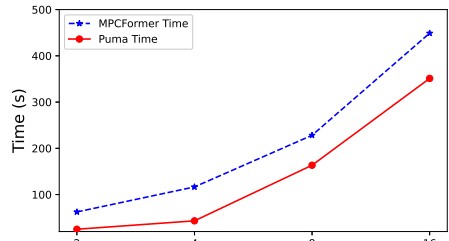

**Output Length Evaluation.** Fig 1 presents our costs on varying-length outputs for GPT2-Base. Our improvements against MPCFORMER range from $1.279 \sim 2.700\times$.

Figure 1: Runtime of GPT2-Base for generating different output tokens, the input length is of length 32.

We observe in Table 5 and Fig 1 that for GPT-2, our efficiency gains decrease with more input/output tokens. This is because PUMA introduces extra one-hot embedding costs (as described in 4.4). We should emphasize again that PUMA is compatible with plaintext models, and could achieve a similar accuracy as plaintext models while MPCFORMER could not.

## 5.4 EVALUATING LLaMA-7B IN FIVE MINUTES.

Our protocols are already complete for evaluating any Transformer-based models including LLaMA-7B. Unfortunately, existing serialization libraries such as Protobuf (Varda, 2008) and FlatBuffers (van Oortmerssen, 2014) only support data trunks with size up to 2GB, which is not sufficient for large MPC tasks. To address this problem, we propose an optimization to SecretFlow-SPU. Concretely, the system could automatically divide and serialize overly large secret-shared structures into smaller chunks when communicating or performing I/O operations.

We evaluated the large language model LLaMA-7B using PUMA under 3 Alibaba Cloud ecs.r7.32xlarge servers, each has 128 threads and 1TB RAM, with 20GB bandwidth, 0.1ms round-trip-time. As shown in Table 6, PUMA can support secure inference of LLaMA-7B with reasonable costs. For example, given an input sentence of 8 tokens, PUMA can output one token in around 200 seconds with communication costs of 1.794 GB. To our knowledge, this is the first time that LLaMA-7B has been evaluated using MPC. Moreover, PUMA can generate the same tokens exactly as plaintext LLaMA-7B, see Appendix for an example.

Table 6: Costs of the secure inference of LLaMA-7B, #Input denotes the length of input sentence and #Output denotes the number of generated tokens.

| (#Input, #Output) | (4, 1) | | (8, 1) | | (8, 2) | |
|---|---|---|---|---|---|---|
| Costs | Time | Comm. | Time | Comm. | Time | Comm. |
| PUMA | 122.004 | 0.907 | 200.473 | 1.794 | 364.527 | 3.857 |

## 6 CONCLUSION

We propose an efficient MPC framework PUMA for secure inference on Transformer models based on replicated secret sharing. To reduce the costs of secure inference, we approximate expensive functions with accurate polynomials and propose secure Embedding and LayerNorm protocols to support end-to-end secure inference. Although the inference cost is still quite high, we successfully make it one step closer to solving users' privacy concerns in Transformer-based DLaaS. We believe that by combining PUMA with quantization methods and hardware accelerations in the future, secure inference of large Transformer models in seconds is no longer impossible.

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

| *Plaintext* | PUMA |
|---|---|
| **Prompt:**
Q: What is the largest animal?
**Outputs:**
A: The largest animal is the blue whale.
Q: What is the smallest animal?
A: The smallest animal is the bee. | **Prompt:**
Q: What is the largest animal?
**Outputs:**
A: The largest animal is the blue whale.
Q: What is the smallest animal?
A: The smallest animal is the bee. |

Figure 2: Outputs of LLaMA-7B in plaintext and PUMA.

## A  DETAILS OF EXPERIMENTAL MODELS

In this section, we present the architecture of the experimental models in brief. For more details, please refer to HuggingFace Transformers library (Wolf et al., 2020).

- Bert-Base: Bert-Base is the base version of the Bert model and consists of 12 Transformer encoder layers, 768 hidden size, and 12 heads. It has 110 million parameters and is trained on a large corpus of unlabeled text data.

- Roberta-Base: Similar to Bert-base, Roberta-base is a base version of the Roberta model. It comprises 12 Transformer layers, 768 hidden size, and 12 heads. It has around 125 million parameters.

- Bert-Large: Bert-Large is an extended version of Bert-base with 24 Transformer encoder layers, 1024 hidden size, and 16 heads. It has approximately 340 million parameters, making it more powerful and capable of capturing complex language patterns.

- GPT2-Base: GPT2-Base is the base version of the Gpt2 model and consists of 12 Transformer decoder layers, 768 hidden size, and 12 heads. It has 117 million parameters and is trained on a large corpus of text data. GPT2-Base is mainly used for tasks involving text generation and language understanding.

- GPT2-Medium: GPT2-Medium comprises 24 Transformer decoder layers, 1024 hidden size, and 16 heads. And it has approximately 345 million parameters.

- GPT2-Large: GPT2-Large is the largest variant of the GPT2 model, featuring 36 Transformer decoder layers, 1280 hidden size, and 16 heads. It has approximately 774 million parameters.

## B  PUMA FOR LLAMA-7B

Unlike GPT-2 and Bert, LLaMA uses SiLU instead of GeLU, we can approximate SiLU using similar piece-wise low-degree polynomials with different coefficients. The full polynomials could be found in $flax\_llama7b.py$ .

In Figure 2, we show the output tokens of LLamA-7B (with fixed randomness) given the prompt: *Q: What is the largest animal?* It can be seen that our PUMA outputs the same tokens as LLaMA-7B does in plaintext for generating more than 20 tokens.

