# OpenReview forum: "PUMA: Secure Inference of LLaMA-7B in Five Minutes"
_ICLR.cc/2024/Conference — Submitted to ICLR 2024_

### Official Review · Reviewer_tTd1 · 2023-10-31

**Soundness:** 3 good
**Presentation:** 4 excellent
**Contribution:** 2 fair
**Rating:** 6
**Confidence:** 3

**Summary:**

This paper brings together a bunch of recent techniques for semi-honest, honest majority 3 server multi party computation and supplements them with a few custom designed gadgets. These are then used to approximate GeLU based neural networks in MPC. They have an experimental section analysing how quickly it runs and the accuracy/precision of the resulting protocol.

**Strengths:**

The paper brings together a bunch of SOTA work well.
It does provide some new approximations, e.g. of GeLU, which seem like useful components for future work.
The paper is well laid out and easy to follow.
The resulting protocol is runnable with fairly large models and gets good approximations.

**Weaknesses:**

The paper seems some what incremental in nature with the new contributions being slightly limited in scope.
It is dubious whether the ability to generate one token from a language model in 5 minutes between three parties in the semi-honest honest majority model is likely to have any applications in the imminent future. But this is a step closer to something practical being possible.

**Questions:**

Are you aware of any plausible near term application that this technology is close to good enough to be deployed for?

---

> ### Author Response · Authors · 2023-11-13
> **Response to Reviewer tTd1**
>
> Thank you for your reviewing efforts and comments.
>
> Yes practical general-purpose privacy-preserving LLMs is like a "holy grail" and hard to achieve, we are still far from that.
> But PUMA does not use any GPUs or quantization methods, so it is very likely that we could benefit from powerful GPUs and ML algorithm optimizations in the future. Once PUMA could be accelarated by one or two magnitudes, it could be possible to support analysis tasks that are not time-sensitive but require private data, e.g., business document or personal chat history processing.

---

### Official Review · Reviewer_bbdZ · 2023-11-01

**Soundness:** 1 poor
**Presentation:** 2 fair
**Contribution:** 1 poor
**Rating:** 3
**Confidence:** 4

**Summary:**

This paper proposes PUMA, a 3PC inference protocol for Transformers.

**Strengths:**

The 3PC setting for LLM is timely.

**Weaknesses:**

1. Limited novelty. The provided protocols are straightforward and contain no novel design or construction.
2. Compared with the baseline, the protocol advantages seem to all come from RSS.
2. Overclaim the experimental performance. The title of this paper is Secure Inference of LLaMA-7B in Five Minutes. However, it is overclaimed. 5 minutes is only when the input and output are 4 and 1 token respectively, which is obviously not in line with the practical service setting.

**Questions:**

See above.

---

> ### Author Response · Authors · 2023-11-13
> **Response to Reviewer bbdZ**
>
> Thank you for your reviewing efforts and comments.
>
> ## For Q1:
> Please check our general response for "lack of novelty".
>
> ## For Q2:
> No, it is not true that "the protocol advantages seem to all come from RSS". RSS-based 3PC could save some communication at the **linear** layers compared to MPCFormer's dealer-mode 3PC protocol, but our major advantages come from the lightweight and accurate **non-linear** approximations. We did an experiment on Llama-7B (8-token input) to replace our GeLU(SiLU) optimization with protocols built on faithful exponent operations (like CrypTen in MPCFormer), and it will increase the overall communication cost drastically (from 1.794GB to 4.303GB).
>
> ## For Q3:
> Describing the cost of LLM in a  "per token" way is a common practice. We argue that a paper should be judged by its merits, rather than whether it's in line with the practical service.

---

### Official Review · Reviewer_DyAw · 2023-11-01

**Soundness:** 2 fair
**Presentation:** 3 good
**Contribution:** 2 fair
**Rating:** 6
**Confidence:** 4

**Summary:**

This work presents a secure Transformer inference framework in 3PC.

**Strengths:**

+ Simple but effective approximations for GELUs.
+ End-to-end framework for Secure LLM Inference.
+ Extensive evaluations.

**Weaknesses:**

The protocols in this work seem limited contributions and are mainly taken from prior works.

**Questions:**

1. What is the cost of secure inference on LLaMA-7B when extending it to a common input length e.g., 128?
2. Does the polynomial approximation of GELU affect the accuracy of large models such as LLaMA-7B because it seems to cause a relatively large error?

---

> ### Author Response · Authors · 2023-11-13
> **Response to Reviewer DyAw**
>
> Thank you for your reviewing efforts and comments.
> ## For Q1:
> We did an experiment running on LLaMA-7B for input with 128 tokens, and the time cost is about 10 minutes. The cost does not grow linearly with the input size (which is also indicated in Table 5).
> ## For Q2:
> Yes even a small perturbation in the activation function could lead to different output. As mentioned in the general response, if we replace our GeLU/SiLU approximation with Quad polynomial approximation like MPCFormer(ICLR23) , the result would be like this:
>
> - Q: What is the largest animal?
> - A: THER grape contend Iraves Grahaminthshireilton ChurchillTHER grape contendinthshireinthshireinthshireinthshireinthshireinthshireinthshireinthshireinth
>
> It serves as an evidence that finding proper approximations is a non-trivial task.

---

### Official Review · Reviewer_G7JD · 2023-11-06

**Soundness:** 3 good
**Presentation:** 3 good
**Contribution:** 2 fair
**Rating:** 5
**Confidence:** 3

**Summary:**

This paper presents a model that utilizes multi-party computation techniques to perform the LLAMA-7B model while preserving the privacy of the client's data. In this process, an optimization of the approximation method for the GeLU function was carried out, and softmax, embedding, and layer normalization methods were all implemented using MPC, achieving an end-to-end implementation. Through these techniques, the computational time has been reduced by approximately 2 times compared to the existing implementation, MPCformer.

**Strengths:**

1. Completion of end-to-end implementation of a large language model using multiparty computation techniques.
2. Achieving inference that is twice as fast as the previously published MPCformer model.
3. The operation of the GeLU function in a different manner compared to the conventional approach.

**Weaknesses:**

1. Most of the methods appear to be simple adaptations of existing techniques, lacking any distinctive novel approach. While the results are impressive from an industrial perspective, it raises doubts about whether they are suitable for ICLR, which places a strong emphasis on academic contributions.

2. While it claims to perform twice as fast as MPCformer, the paper lacks precise explanations of why each technique is superior to the existing ones, making it challenging to assess their effectiveness.

3. The primary technical contribution seems to be the approximation of the GELU function. However, without a clear comparison to existing approximations, it is challenging to assess the value of this technique. The paper introduces variations in polynomial computation based on the range of x, but it is not evident how this approach is superior to the conventional method of approximating the GELU function in terms of computational efficiency.

4. Regarding the meaningful academic contributions in softmax, embedding, and layerwise normalization, it is not clear where they lie, making it difficult to discern the significance of these contributions.

**Questions:**

1. Provide numerical evidence of how the computations involved in our method for approximating the GELU function offer advantages in terms of computational and communication costs compared to existing techniques that achieve the same level of accuracy.

2. Convince that the techniques employed in softmax, embedding, and layerwise normalization go beyond mere combinations of existing methods and provide non-trivial technical contributions.

3. Explain what specific factors contributed to the 2x performance improvement compared to MPCformer and quantitatively specify the performance gains achieved by each factor.

---

> ### Author Response · Authors · 2023-11-13
> **Response to Reviewer G7JD**
>
> Thank you for your reviewing efforts and comments.
>
> ## For Q1 and Q3:
>
> There are many optimization details under PUMA, roughly speaking, GeLU and softmax contribute to the efficiency gains, while embedding, and layernorm are vital to usability. In fact embedding and layernorm decrease efficiency because they are absent in previous works, but we cannot output meaningful results without them. The potion of cost for each module varies with different models. Due to the space limit, we cannot afford to discuss these costs one by one, but GeLU usually contributes the major part in efficiency gains.
>
> We did an experiment on Llama-7B (8-token input) to replace our GeLU(SiLU) optimization with protocols built on faithful exponent operations (like CrypTen in MPCFormer), and it will increase the overall communication drastically (from 1.794GB to 4.303GB).
>
> ## For Q2:
> Bringing all the simple and efficient submodules altogether to achieve an end-to-end solution is a non-trivial task. An example is [R1]: it is also made of several building blocks built from existing works.  Each building block might seem to be "incremental", but it does not affect the overall contribution. In fact, it serves as the state-of-the-art in the area of MPC machine learning training.
>
> _[R1] Keller M, Sun K. Secure quantized training for deep learning[C]//International Conference on Machine Learning. PMLR, 2022: 10912-10938. (ICML22 spotlight)_

---

### Author Response · Authors · 2023-11-13
**General Response for "lack of novelty"**

We thank all the reviewers for their efforts and insightful comments. We respond to the general concerns for "lack of novelty" below, and respond to each reviewer for individual questions.

Our main goal (and major novelty) is "Running LLM under MPC without dropping accuracy", which includes several optimized protocols such as GeLU, Softmax, embedding, and layernorm, and the efforts of bringing all together.

We argue that novel approaches don't have to be complex. Finding simple and efficient solutions and bringing them altogether could also be a contribution. Previous works failed to achieve that. An evidence is that before we conduct research on PUMA, there do not exist any **runnable** MPC solutions that can output meaningful results on LLM, even for the much simpler GPT-2 (which has been proposed for four years!).

If we replace our GeLU/SiLU approximation with Quad approximation like MPCFormer(ICLR23) , the result would be like this:

- Q: What is the largest animal?
- A: THER grape contend Iraves Grahaminthshireilton ChurchillTHER grape contendinthshireinthshireinthshireinthshireinthshireinthshireinthshireinthshireinth

If we replace our GeLU/SiLU approximation with ReLU approximation like PrivFormer(EuroSP23), the result would be like this:
- Q: What is the largest animal?
- A: * a. What is the difference between? Sedition between? Sedition ? ? between

On the contrary, PUMA not only output meaningful results, but also achieves exactly the same results as plaintext LLaMA-7B.

---

### Meta-Review · Area_Chair_aWgF · 2023-12-10

**Metareview:**

The reviewers were split about this paper and did not come to a consensus: on one hand they appreciated the inference speedup compared to prior work and the new GeLU approximation, on the other they had issues with (a) missing comparisons with related work and ablation studies, and (b) limited novelty. After going through the paper and the discussion I have decided to vote to reject based on the above issues. Specifically for (a), the reviewers wanted to see comparisons between the new GeLU approximation and other prior work. Further, they wanted to understand what specific factors contributed to the 2x time improvement over MPCformer. The authors did provide accuracy, time, and communication comparisons for the overall model, but did not give a specific comparison on the GeLU approximation with other approximation or provide additional insight into what specifically contributes to the 2x time improvement. As such, this point is not fully resolved. For (b), the reviewers argued that the paper is largely a combination of prior techniques (apart from the GeLU approximation). The authors responded that while prior work ensures security it does not result in an LLM that outputs meaningful responses to prompts, and they show the responses to an example prompt to demonstrate this. This response is useful but inadequate, the authors should have also reported the full performance of baselines on GLUE and Wikitext as they did for their model (i.e., Tables 1 and 2). It’s unclear if the prompt they chose to report in the rebuttal is cherry-picked to make these baselines look bad, and the full performance results would have resolved this. Beyond these two points, the writing is imprecise in multiple places. For instance, the paper says “we compare our performance to that of plaintext (floating-point) to show our precision guarantee”. But there is no such guarantee (i.e., no formal statement describing what precision one would expect to see). Given all of the above, I believe this work should be rejected at this time. Once these things and other issues mentioned in the reviews are addressed in an updated version, the work will be much improved.

**Justification For Why Not Higher Score:**

The authors never adequately responded to the reviewer feedback, leaving serious concerns open.

**Justification For Why Not Lower Score:**

N/A

---

### Decision · Program_Chairs · 2024-01-16

Reject